# STABILITY AND GENERALIZATION IN FREE ADVERSARIAL TRAINING

## ABSTRACT

While adversarial training methods have resulted in significant improvements in the deep neural nets' robustness against norm-bounded adversarial perturbations, their generalization performance from training samples to test data has been shown to be considerably worse than standard empirical risk minimization methods. Several recent studies seek to connect the generalization behavior of adversarially trained classifiers to various gradient-based min-max optimization algorithms used for their training. In this work, we study the generalization performance of adversarial training methods using the algorithmic stability framework. Specifically, our goal is to compare the generalization performance of vanilla adversarial training scheme fully optimizing the perturbations at every iteration vs. the free adversarial training simultaneously optimizing the norm-bounded perturbations and classifier parameters. Our proven generalization bounds indicate that the free adversarial training method could enjoy a lower generalization gap between training and test samples due to the simultaneous nature of its min-max optimization algorithm. We perform several numerical experiments to evaluate the generalization performance of vanilla, fast, and free adversarial training methods. Our empirical findings also show the improved generalization performance of the free adversarial training method and further demonstrate that the better generalization result could translate to greater robustness against black-box attack schemes and higher transferability of the adversarial examples designed for free adversarially trained neural networks.

## 1 INTRODUCTION

While deep neural networks (DNNs) have led to remarkable results in standard supervised learning tasks in computer vision and natural language processing, they are widely recognized to be susceptible to minor adversarially-designed perturbations to their input data commonly regarded as *adversarial attacks* (Szegedy et al., 2013; Goodfellow et al., 2014). Adversarial examples are typically designed by finding the worst-case norm-constrained perturbation that leads to the maximum impact on the classification loss at an input data point. To combat norm-bounded adversarial attacks, adversarial training (AT) methods (Madry et al., 2017) which learn a DNN classifier using adversarially-perturbed training examples have been shown to significantly improve the robustness of a DNN against norm-bounded adversarial attacks. Several variants of AT methods have been developed in the machine learning community to accelerate and facilitate the application of AT algorithms to large-scale machine learning problems (Shafahi et al., 2019; Wong et al., 2020).

While AT algorithms have achieved state-of-the-art robustness scores against standard norm-bounded adversarial attacks, the generalization gap between their performance on training and test data has been frequently observed to be significantly greater than the generalization error of DNNs learned by standard empirical risk minimization (ERM) (Schmidt et al., 2018; Raghunathan et al., 2019). To understand the significant generalization gap in adversarial training, several theoretical and empirical studies have focused on the generalization properties of adversarially-trained models (Yin et al., 2019; Rice et al., 2020). These studies have attempted to analyze the generalization error in learning adversarially-robust models and reduce the generalization gap by applying explicit and implicit regularization techniques such as early stopping and Lipshcitz regularization methods.

Specifically, several recent works (Lei et al., 2021; Farnia & Ozdaglar, 2021; Xiao et al., 2022b) have focused on the connections between the optimization and generalization behavior of adversarially-trained models. Since adversarial training methods use adversarial training examples with the worst-case norm-bounded perturbations, they are typically formulated as min-max optimization problems where the classifier and adversarial perturbations are the minimization and maximization variables, respectively. To solve the min-max optimization problem, the vanilla AT framework follows an iterative algorithm where, at every iteration, the inner maximization problem is fully solved for designing the optimal perturbations and subsequently, a single gradient update is applied to the DNN's parameters. Therefore, the vanilla AT results in a non-simultaneous optimization of the minimization and maximization variables of the underlying min-max problem. However, the theoretical generalization error bounds in (Farnia & Ozdaglar, 2021; Lei et al., 2021) suggest that the non-simultaneous optimization of the min and max variables in a min-max learning problem could lead to a greater generalization gap. Therefore, a natural question is whether an adversarial training algorithm with simultaneous optimization of the min and max problems can reduce the generalization gap.

In this work, we focus on a widely-used variant of adversarial training proposed by Shafahi et al. (2019), *adversarial training for free (free AT)*, and aim to analyze its generalization behavior compared to the vanilla AT approach. While the vanilla AT follows a sequential optimization of the DNN and perturbation variables, the Free AT approach simultaneously computes the gradient of the two groups of variables at every round of applying the backpropagation algorithm to the multi-layer DNN. We aim to demonstrate that the mentioned simultaneous optimization of the classifier and adversarial examples in free AT could translate into a lower generalization error compared to vanilla AT. To this end, we provide theoretical and numerical results to compare the generalization properties of vanilla vs. free AT frameworks.

On the theory side, we leverage the algorithmic stability framework (Bousquet & Elisseeff, 2002; Hardt et al., 2015) to derive generalization error bounds for free and vanilla adversarial training methods. The shown generalization bounds suggest that in the nonconvex-nonconcave regime, the free AT algorithm could enjoy a lower generalization gap than the vanilla AT, since it applies simultaneous gradient updates to the DNN's and perturbations' variables. We also develop a similar generalization bound for the fast AT methodology (Goodfellow et al., 2014) which uses a single gradient step to optimize the perturbations. Our theoretical results suggest a comparable generalization bound between free and fast AT approaches.

Finally, we present the results of our numerical experiments to compare the generalization performance of the vanilla, fast, and free AT methods over standard computer vision datasets and neural network architectures. Our numerical results also suggest that the free AT method results in a considerably lower generalization gap than the vanilla AT and relatively improves the generalization performance over the fast AT algorithm. While the lower generalization error of free AT does not lead to a significant improvement of the test accuracy under white-box PGD attacks, our empirical results suggest that the networks trained by free AT result in a higher test accuracy under standard black-box adversarial attacks. Furthermore, our numerical findings indicate that the adversarial perturbations designed for DNNs trained by free AT could transfer better to an unseen target neural net classifier than those optimized for DNNs trained according to vanilla and fast AT. We can summarize this work's contributions as follows:

- Leveraging the algorithmic stability framework to analyze the generalization behavior of the free AT algorithm,

- Providing a theoretical comparison of the generalization properties of the vanilla, fast, and free AT methods,

- Numerically analyzing the generalization and test performance of the free vs. vanilla AT schemes under white-box and black-box adversarial attacks.

## 2 RELATED WORK

**Generalization in Adversarial Training:** Since the discovery of adversarial examples (Szegedy et al., 2013), a large body of works has focused on training robust DNNs against adversarial perturbations (Goodfellow et al., 2014; Carlini & Wagner, 2017; Madry et al., 2017; Zhang et al., 2019). Shafahi et al. (2019) proposed "free" adversarial training algorithm to update the neural net and

adversarial perturbations simultaneously, and Wong et al. (2020) proposed "fast" algorithm, both of which were originally aimed at reducing the computational cost of adversarial training. Compared to standard training, the overfitting in adversarial training is shown to be significantly more severe (Rice et al., 2020). A line of works analyzed adversarial generalization through the lens of uniform convergence analysis such as via VC-dimension (Montasser et al., 2019; Attias et al., 2022) and Rademacher complexity (Yin et al., 2019; Farnia et al., 2018; Awasthi et al., 2020; Xiao et al., 2022a). Schmidt et al. (2018) proved tight bounds on the adversarially robust generalization error showing that vanilla adversarial training requires more data for proper generalization than standard training. Xing et al. (2022) studied the phase transition of generalization error from standard training to adversarial training. Also, the reference (Andriushchenko & Flammarion, 2020) discusses the catastrophic overfitting in the Fast AT method.

**Uniform Stability:** Bousquet & Elisseeff (2002) developed the algorithmic stability framework to analyze the generalization performance of learning algorithms. Hardt et al. (2015) further extended the algorithmic stability approach to stochastic gradient-based optimization (SGD) methods. Bassily et al. (2020); Lei (2023) analyzed the stability under non-smooth functions. Some recent works applied the stability framework to study the generalization gap of adversarial training, while they mostly assumed an oracle to obtain a perfect perturbation and focused on the stability of the training process. Xing et al. (2021) analyzed the stability by shedding light on the non-smooth nature of the adversarial loss. Xiao et al. (2022b) further investigated the stability bound by introducing a notion of approximate smoothness. Based on this result, Xiao et al. (2022c) proposed a smoothed version of SGDmax to improve the adversarial generalization. Xiao et al. (2023) utilized the stability framework to improve the robustness of DNNs under various types of attacks.

**Generalization in minimax learning frameworks:** The generalization analysis of general minimax learning frameworks has been studied in several related works. Arora et al. (2017) established a uniform convergence generalization bound in terms of the discriminator's parameters in generative adversarial networks (GANs). Zhang et al. (2017); Bai et al. (2018) characterized the generalizability of GANs using the Rademacher complexity of the discriminator function space. Some work also analyzed generalization in GANs from the algorithmic perspective. Farnia & Ozdaglar (2021); Lei et al. (2021) compared the generalization of SGDA and SGDmax in minimax optimization problems using algorithmic stability. Wu et al. (2019) studied generalization in GANs from the perspective of differential privacy. Ozdaglar et al. (2022) proposed a new metric to evaluate the generalization of minimax problems and studied the generalization behaviors of SGDA and SGDmax.

## 3 PRELIMINARIES

Suppose that labelled sample $(x, y)$ is randomly drawn from some unknown distribution $\mathcal{D}$. The goal of adversarial training is to find a model $f_w$ with parameter $w \in W$ which minimizes the population risk against the adversarial perturbation $\delta$ from a feasible perturbation set $\Delta$, defined as:

$$R(w) := \mathbb{E}_{(x,y) \sim \mathcal{D}} \left[ \max_{\delta \in \Delta} h(w, \delta; x, y) \right],$$

where $h(w, \delta; x, y) = \text{Loss}(f_w(x + \delta), y)$ is the loss function in the supervised learning problem. Since the learner does not have access to the underlying distribution $\mathcal{D}$ but only a dataset $S = \{x_1, x_2, \cdots, x_n\}$ of size $n$, we define the empirical adversarial risk as

$$R_S(w) := \frac{1}{n} \sum_{j=1}^{n} \max_{\delta \in \Delta} h(w, \delta; x_j, y_j).$$

The generalization adversarial risk $\mathcal{E}_{\text{gen}}(w)$ of model parameter $w$ is defined as the difference between population and empirical risk, i.e., $\mathcal{E}_{\text{gen}}(w) := R(w) - R_S(w)$. For a potentially randomized algorithm $A$ which takes a dataset $S$ as input and outputs a random vector $w = A(S)$, we can define its expected generalization adversarial risk over the randomness of a training set $S$ and stochastic algorithm $A$, e.g. under mini-batch selection in stochastic gradient methods or random initialization of the weights of a neural net classifier,

$$\mathcal{E}_{\text{gen}}(A) := \mathbb{E}_{S,A} \big[ R(A(S)) - R_S(A(S)) \big].$$

Throughout the paper, unless specified otherwise, we use $\| \cdot \|$ to denote the $\mathcal{L}_2$ norm of vectors or the Frobenius norm of matrices.

### 3.1 ADVERSARIAL TRAINING

In the field of adversarial training, the perturbation set $\Delta$ is usually an $\mathcal{L}_2$-norm or $\mathcal{L}_\infty$-norm bounded ball of some small radius $\varepsilon$ (Szegedy et al., 2013; Goodfellow et al., 2014). To robustify a neural network, the standard methodology $A_{\text{Vanilla}}$ is to train the network with (approximately) perfectly perturbed samples, both in practice (Madry et al., 2017; Rice et al., 2020) and in theory analysis (Xing et al., 2021; Xiao et al., 2022b), which is formally defined as follows:

---

**Algorithm 1** Vanilla Adversarial Training Algorithm $A_{\text{Vanilla}}$

---

1: **Input:** Training samples $S$, perturbation set $\Delta$, learning rate of model weight $\alpha_w$, mini-batch size $b$, number of iterations $T$
2: **for** step $t \leftarrow 1, \cdots, T$ **do**
3:     Uniformly random mini-batch $B \subset S$ of size $b$
4:     Compute adversarial attack $\delta_j$ for all $(x_j, y_j) \in B$: $\delta_j \leftarrow \arg\max_{\tilde{\delta} \in \Delta} h(w, \tilde{\delta}; x_j, y_j)$
5:     Update $w$ with perturbed samples: $w \leftarrow w - \frac{\alpha_w}{b} \sum_{(x_j, y_j) \in B} \nabla_w h(w, \delta_j; x_j, y_j)$
6: **end for**

---

In practice, due to the non-convexity of neural networks, it is computationally intractable to compute the best adversarial attack $\delta = \arg\max_{\tilde{\delta} \in \Delta} h(w, \tilde{\delta}; x)$, but the standard projected gradient descent (PGD) attack (Madry et al., 2017) is widely believed to produce near-optimal attacks, by iteratively projecting the gradient $\nabla_\delta h(w, \delta; x)$ onto the set of extreme points of $\Delta$, i.e.,

$$\pi_\Delta(g) := \underset{\tilde{\delta} \in \text{ExtremePoints}(\Delta)}{\arg\min} \|g - \tilde{\delta}\|^2, \tag{1}$$

updating the attack $\delta$ with the projected gradient $\pi_\Delta(\nabla_\delta h(w, \delta; x))$ and some step size $\alpha_\delta$, and projecting the update attack to the feasible set $\Delta$, i.e,

$$\mathcal{P}_\Delta(g) := \underset{\delta \in \Delta}{\arg\min} \|g - \delta\|^2. \tag{2}$$

Despite the significant robustness gained from $A_{\text{Vanilla}}$, it demands high computational costs for training. The "free" adversarial training algorithm $A_{\text{Free}}$ (Shafahi et al., 2019) is proposed to avoid the overhead cost, by simultaneously updating the model weight parameter $w$ when performing PGD attacks. $A_{\text{Free}}$ is empirically observed to achieve comparable robustness to $A_{\text{Vanilla}}$, while it can considerably reduce the training time (Shafahi et al., 2019; Wong et al., 2020).

---

**Algorithm 2** Free Adversarial Training Algorithm $A_{\text{Free}}$

---

1: **Input:** Training samples $S$, perturbation set $\Delta$, step size of model weight $\alpha_w$, learning rate of adversarial attack $\alpha_\delta$, free step $m$, mini-batch size $b$, number of iterations $T$
2: **for** step $\leftarrow 1, \cdots, T/m$ **do**
3:     Uniformly random mini-batch $B \subset S$ of size $b$
4:     $\delta := [\delta_j]_{\{j:x_j, y_j \in B\}} \leftarrow \text{Uniform}(\Delta^b)$
5:     **for** iteration $i \leftarrow 1, \cdots, m$ **do**
6:         Compute weight gradient and attack gradient by backpropagation:
7:         $g_w \leftarrow \frac{1}{b} \sum_{x_j, y_j \in B} \nabla_w h(w, \delta_j; x_j, y_j)$, and $g_\delta \leftarrow [\nabla_\delta h(w, \delta_j; x_j, y_j)]_{\{j:x_j, y_j \in B\}}$
8:         Update $w$ with mini-batch gradient descent: $w \leftarrow w - \alpha_w g_w$
9:         Update $\delta$ with projected gradient ascent: $\delta \leftarrow [\mathcal{P}_\Delta(\delta_j + \alpha_\delta \pi_\Delta(g_{\delta_j}))]_{\{j:x_j, y_j \in B\}}$
10:     **end for**
11: **end for**

---

We also compare $A_{\text{Vanilla}}$ and $A_{\text{Free}}$ with the "fast" adversarial training algorithm $A_{\text{Fast}}$ (Wong et al., 2020), which is a variant of the fast gradient sign method (FGSM) by Goodfellow et al. (2014). Instead of computing a perfect perturbation, it applies only one projected gradient step with fine-tuned step size from a randomly initialized point in $\Delta$. It is also empirically observed to achieve comparable robustness with fewer cost (Wong et al., 2020; Andriushchenko & Flammarion, 2020).

---

**Algorithm 3** Fast Adversarial Training Algorithm $A_{\text{Fast}}$

---

1: **Input:** Training samples $X$, perturbation set $\Delta$, , learning rate of model weight $\alpha_w$, step size of adversarial attack $\tilde{\alpha}_\delta$, mini-batch size $b$, number of iterations $T$
2: **for** step $t \leftarrow 1, \cdots, T$ **do**
3:     Uniformly random mini-batch $B \subset S$ of size $b$
4:     Compute adversarial attack $\delta$ with random start:
5:     $\tilde{\delta} := [\tilde{\delta}_j]_{\{j:x_j,y_j \in B\}} \leftarrow \text{Uniform}(\Delta^b)$
6:     $g_\delta \leftarrow [\nabla_\delta h(w, \tilde{\delta}_j; x_j, y_j)]_{\{j:x_j,y_j \in B\}}$
7:     $\delta \leftarrow [\mathcal{P}_\Delta(\tilde{\delta}_j + \alpha_\delta \pi_\Delta(g_{\delta_j}))]_{\{j:x_j,y_j \in B\}}$
8:     Update $w$ with perturbed sample: $w \leftarrow w - \frac{\alpha_w}{b} \sum_{x_j,y_j \in B} \nabla_w h(w, \delta_j; x_j, y_j)$
9: **end for**

---

## 4 STABILITY AND GENERALIZATION IN ADVERSARIAL TRAINING

To bound the generalization adversarial risk, the notion of uniform stability with respect to the adversarial loss is introduced (Bousquet & Elisseeff, 2002).

**Definition 1.** *A randomized algorithm $A$ is $\epsilon$-uniformly stable if for all datasets $S, S' \in \mathcal{D}^n$ such that $S$ and $S'$ differ in at most one example, we have*

$$\sup_x \mathbb{E}_A \left[ \max_{\delta \in \Delta} h(A(S), \delta; x) - \max_{\delta \in \Delta} h(A(S'), \delta; x) \right] \leq \epsilon. \tag{3}$$

As Theorem 2.2 in Hardt et al. (2015), the generalization risk in expectation of a uniformly stable algorithm can be bounded by the following theorem

**Theorem 1.** *Assume that a randomized algorithm $A$ is $\epsilon$-uniformly stable, then the expected generalization risk satisfies*

$$|\mathcal{E}_{gen}| = |\mathbb{E}_{S,A}[R(A(S)) - R_S(A(S))]| \leq \epsilon.$$

*Proof.* The proof can be found in Theorem 2.2 in Hardt et al. (2015) by replacing the loss function with the adversarial loss $\max_{\delta \in \Delta} h(w, \delta; x)$. $\square$

In order to study the uniform stability of adversarial training, we make the following assumptions on the Lipschitzness and smoothness of the objective function. Our generalization results will hold as long as Assumptions 1, 2 hold locally within an attack radius distance from the support set of $X$.

**Assumption 1.** *$h(w, \delta)$ is jointly $L$-Lipschitz in $(w, \delta)$ and $L_w$-Lipschitz in $w$ over $W \times \Delta$, i.e., for every $w, w' \in W$ and $\delta, \delta' \in \Delta$ we have*

$$|h(w, \delta) - h(w', \delta')|^2 \leq L^2 \left( \|w - w'\|^2 + \|\delta - \delta'\|^2 \right), \quad |h(w, \delta) - h(w', \delta)|^2 \leq L_w^2 \|w - w'\|^2.$$

**Assumption 2.** *$h(w, \delta)$ is continuously differentiable and $\beta$-smooth over $W \times \Delta$, i.e., $[\nabla_w h(w, \delta), \nabla_\delta h(w, \delta)]$ is $\beta$-Lipschitz over $W \times \Delta$ and for every $w, w' \in W$, $\delta, \delta' \in \Delta$ we have*

$$\|\nabla_w h(w, \delta) - \nabla_w h(w', \delta')\|^2 + \|\nabla_\delta h(w, \delta) - \nabla_\delta h(w', \delta')\|^2 \leq \beta^2 \left( \|w - w'\|^2 + \|\delta - \delta'\|^2 \right).$$

We clarify that the Lipschitzness and smoothness assumptions are common practice in the uniform stability analysis (Hardt et al., 2015; Xing et al., 2021; Farnia & Ozdaglar, 2021; Xiao et al., 2022b). In practice, although ReLU activation function is non-smooth, recent works (Du et al., 2019; Allen-Zhu et al., 2019) showed that the loss function of over-parameterized neural networks is semi-smooth; also, another line of works (Xie et al., 2020; Singla et al., 2021) suggest that replacing ReLU with smooth activation functions can strengthen adversarial training; and some works (Fazlyab et al., 2019; Shi et al., 2022) attempt to compute the Lipschitz constant of neural networks.

## 5 STABILITY-BASED GENERALIZATION BOUNDS FOR FREE AT

In this section, we provide generalization bounds on vanilla, fast, and free adversarial training algorithms. While previous works mainly focus on theoretically analyzing the stability behaviors of

vanilla adversarial training under the scenario that $h(w, \delta; x)$ is convex in $w$ (Xing et al., 2021; Xiao et al., 2022b), or $h(w, \delta; x)$ is concave or even strongly-concave in $\delta$(Lei et al., 2021; Farnia & Ozdaglar, 2021; Yang et al., 2022; Ozdaglar et al., 2022), our analysis focuses on the nonconvex-nonconcave scenario: without assumptions on the convexity of $h(w, \delta; x)$ in $w$ or concavity of $h(w, \delta; x)$ in $\delta$. We defer the proof of Theorems 2 and 4 to the Appendix A.1 and A.2. Throughout the proof, we assume that Assumptions 1 and 2 hold.

**Theorem 2** (Stability generalization bound of $A_{\text{Vanilla}}$). *Assume that $h(w, \delta)$ satisfies Assumptions 1 and 2 and is bounded in $[0, 1]$, and the perturbation set is an $\mathcal{L}_2$-norm ball of some constant radius $\varepsilon$, i.e., $\Delta = \{\delta : ||\delta|| \leq \varepsilon\}$. Suppose that we run $A_{Vanilla}$ in Algorithm 1 for $T$ steps with vanishing step size $\alpha_{w,t} \leq c/t$. Letting constant $\lambda_{Vanilla} := \beta c$, then*

$$\mathcal{E}_{gen}(A_{Vanilla}) \leq \frac{b}{n}\left(1 + \frac{1}{\lambda_{Vanilla}}\right)\left(\frac{2L_w c}{b}(\varepsilon\beta n + L)\right)^{\frac{1}{\lambda_{Vanilla}+1}} T^{\frac{\lambda_{Vanilla}}{\lambda_{Vanilla}+1}}. \tag{4}$$

By equation 4, we have the following asymptotic bound on $\mathcal{E}_{\text{gen}}(A_{\text{Vanilla}})$ with respect to $T$ and $n$

$$\mathcal{E}_{gen}(A_{Vanilla}) = \mathcal{O}\left(T^{\frac{\lambda_{Vanilla}}{\lambda_{Vanilla}+1}} / n^{\frac{\lambda_{Vanilla}}{\lambda_{Vanilla}+1}}\right). \tag{5}$$

This bound suggests that the vanilla adversarial training algorithm could lead to large generalization gaps, because for any $T = \Omega(n)$, the bound $T^{\frac{\lambda_{Vanilla}}{\lambda_{Vanilla}+1}} / n^{\frac{\lambda_{Vanilla}}{\lambda_{Vanilla}+1}} = \Omega(1)$ is non-vanishing even when we are given infinity samples. This implication is also confirmed by the following lower bound from the work of Xing et al. (2021) and Xiao et al. (2022b):

**Theorem 3** (Lower bound on stability; Theorem 1 in Xing et al. (2021), Theorem 5.2 in Xiao et al. (2022b)). *Suppose $\Delta = \{\delta : ||\delta|| \leq \varepsilon\}$. Assume $w(S)$ is the output of running $A_{Vanilla}$ on the dataset $S$ with mini-batch size $b = 1$ and constant step size $\alpha_w \leq 1/\beta$ for $T$ steps. There exist some loss function $h(w, \delta; x)$ which is differentiable and convex with respect to $w$, some constant $\varepsilon > 0$, and some datasets $S$ and $S'$ that differ in only one sample, such that*

$$\mathbb{E}[||w(S) - w(S')||] \geq \Omega\left(\sqrt{T} + \frac{T}{n}\right). \tag{6}$$

This lower bound indicates that $A_{\text{Vanilla}}$ could lack stability when the attack radius $\varepsilon = \Omega(1)$, hence the algorithm may result in significant generalization error from the stability perspective. Note that the lower bound in equation 6 is not inconsistent with Theorem 2, in which the step-size is assumed to be vanishing $\alpha_{w,t} \leq c/t$ and thus the lower bound is not directly applicable under that assumption. However, this constant generalization gap could be reduced by free adversarial training.

**Theorem 4** (Stability generalization bound of $A_{\text{Free}}$). *Assume that $h(w, \delta)$ satisfies Assumptions 1 and 2 and is bounded in $[0, 1]$, and the perturbation set is an $\mathcal{L}_2$-norm ball of some constant radius $\varepsilon$, i.e., $\Delta = \{\delta : ||\delta|| \leq \varepsilon\}$. Suppose that we run $A_{Free}$ in Algorithm 2 for $T/m$ steps with vanishing step size $\alpha_{w,t} \leq c/mt$ and constant step size $\alpha_\delta$. If the norm of gradient $\nabla_\delta h(w, \delta; x)$ is lower bounded by $1/\psi$ for some constant $\psi > 0$ with probability 1 during the training process, letting constant $\lambda_{Free} := \beta c(1 + \beta c/m + \alpha_\delta \varepsilon \psi \beta)^{m-1}$, then*

$$\mathcal{E}_{gen}(A_{Free}) \leq \frac{b}{n}\left(1 + \frac{1}{\lambda_{Free}}\right)\left(\frac{2LL_w}{b\beta}\lambda_{Free}\right)^{\frac{1}{\lambda_{Free}+1}}\left(\frac{T}{m}\right)^{\frac{\lambda_{Free}}{\lambda_{Free}+1}}. \tag{7}$$

**Remark 1.** *Theorem 4 indicates how the simultaneous updates influence the generalization of adversarial training. From equation 7, we have the following asymptotic bound on $\mathcal{E}_{gen}(A_{Free})$ with respect to $T$ and $n$*

$$\mathcal{E}_{gen}(A_{Free}) = \mathcal{O}\left(T^{\frac{\lambda_{Free}}{\lambda_{Free}+1}} / n\right). \tag{8}$$

*Therefore, by controlling the step size $\alpha_\delta$ of the maximization step, we can bound the coefficient $\lambda_{Free}$ and thus control the generalization gap of $A_{Free}$, where a lower $\alpha_\delta$ can result in a smaller generalization gap.*

Comparing equation 8 with equation 5 suggests that for any $T = \mathcal{O}(n)$, $A_{\text{Free}}$ can generalize better than $A_{\text{Vanilla}}$, since

$$\frac{T^{\frac{\lambda_{Free}}{\lambda_{Free}+1}} / n}{T^{\frac{\lambda_{Vanilla}}{\lambda_{Vanilla}+1}} / n^{\frac{\lambda_{Vanilla}}{\lambda_{Vanilla}+1}}} = \left(\frac{T}{n}\right)^{\frac{1}{\lambda_{Vanilla}+1}}\left(\frac{1}{T}\right)^{\frac{1}{\lambda_{Free}+1}} = \mathcal{O}\left(1/T^{\frac{1}{\lambda_{Free}+1}}\right).$$

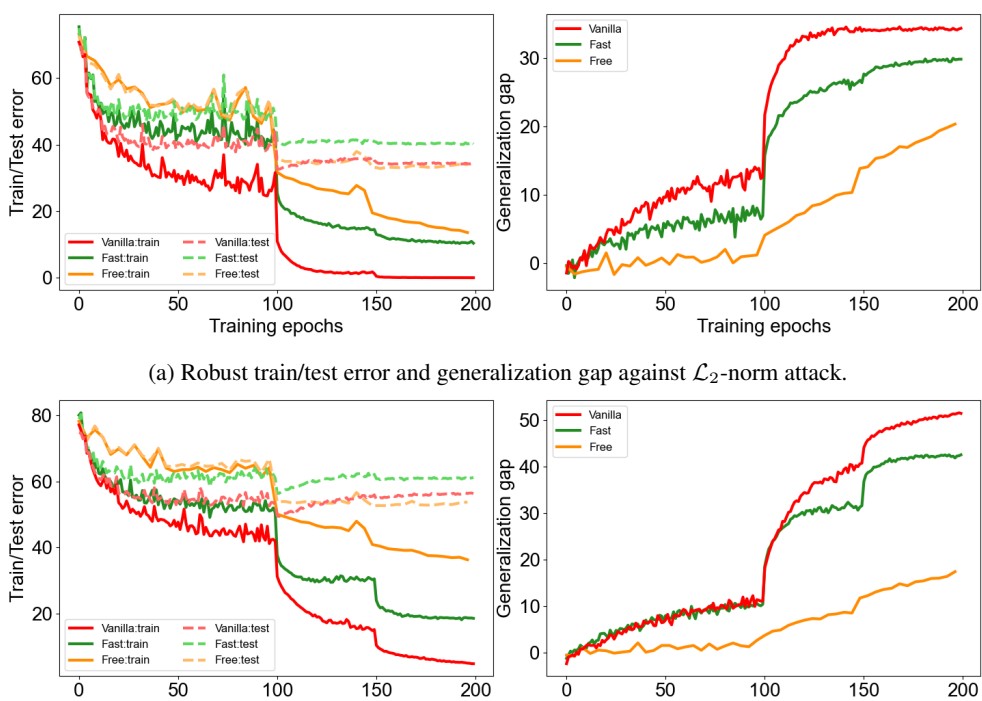

(a) Robust train/test error and generalization gap against $\mathcal{L}_2$-norm attack.

(b) Robust train/test error and generalization gap against $\mathcal{L}_\infty$-norm attack.

Figure 1: Learning curves of different algorithms for a ResNet18 model adversarially trained against $\mathcal{L}_2$ and $\mathcal{L}_\infty$ attacks on CIFAR-10. The free curves are scaled horizontally by a factor of $m$.

Furthermore, when $T = \mathcal{O}(n)$, equation 8 gives $\mathcal{E}_{\text{gen}}(A_{\text{Free}}) = \mathcal{O}\left(1/n^{\frac{1}{\lambda_{\text{Free}}+1}}\right)$, which implies that the generalization gap of $A_{\text{Free}}$ can be bounded given enough samples. If the number of iterations $T$ is fixed, one can see that the generalization gap of $A_{\text{Free}}$ has a faster convergence to 0 than $A_{\text{Vanilla}}$. Therefore, neural nets trained by the free adversarial algorithm could generalize better than the vanilla adversarially-trained networks due to their improved algorithmic stability. Our theoretical results also echo the conclusion in Schmidt et al. (2018) that adversarially robust generalization requires more data, since $\lambda_{\text{Free}}$ increases with respect to $\varepsilon$. We also provide theoretical analysis for the fast adversarial training algorithm $A_{\text{Fast}}$ in Appendix A.3.

## 6 NUMERICAL RESULTS

In this section, we evaluate the generalization performance of vanilla, fast, and free adversarial training algorithms in a series of numerical experiments. We first demonstrate the overfitting issue in vanilla adversarial training and show that free or fast algorithms can considerably reduce the generalization gap. We demonstrate that the smaller generalization gap could translate into greater robustness against score-based and transferred black-box attacks. To examine the advantages of free AT, we also study the generalization gap for different numbers of training samples.

**Experiment Settings:** We conduct our experiments on datasets CIFAR-10, CIFAR-100 (Krizhevsky & Hinton, 2009), Tiny-ImageNet (Le & Yang, 2015), and SVHN (Netzer et al., 2011). Following the standard setting in Madry et al. (2017), we use ResNet18 (He et al., 2016) for CIFAR-10 and CIFAR-100, ResNet50 for Tiny-ImageNet, and VGG19 (Simonyan & Zisserman, 2014) for SVHN to validate our results on a diverse selection of network architectures. For vanilla adversarial training algorithm, since the inner optimization task $\max_{\delta \in \Delta} h(w, \delta; x)$ is computationally intractable for neural networks which are generally non-concave, we apply standard projected gradient descent (PGD) attacks (Madry et al., 2017) as a surrogate adversary. For free and fast algorithms, we adopt $A_{\text{Free}}$ and $A_{\text{Fast}}$ defined in Algorithms 2 and 3, following Shafahi et al. (2019); Wong et al. (2020).

Table 1: Robust generalization performance of different algorithms for a ResNet18 model adversarially trained against $\mathcal{L}_2$-norm and $\mathcal{L}_\infty$-norm attacks on CIFAR-10. We run five independent trials and report the mean and standard deviation of the robust accuracy on training and testing datasets.

| Results (%) | $\mathcal{L}_2$-norm attack | | | $\mathcal{L}_\infty$-norm attack | | |
|---|---|---|---|---|---|---|
| | Vanilla | Fast | Free | Vanilla | Fast | Free |
| Train Acc. | $100.0 \pm 0.0$ | $89.6 \pm 0.2$ | $86.5 \pm 0.2$ | $95.0 \pm 0.3$ | $81.3 \pm 0.4$ | $63.6 \pm 0.3$ |
| Test Acc. | $65.5 \pm 0.2$ | $59.8 \pm 0.4$ | $65.7 \pm 0.4$ | $43.8 \pm 0.1$ | $39.1 \pm 0.2$ | $46.4 \pm 0.3$ |
| Gen. Gap | $34.5 \pm 0.2$ | $29.8 \pm 0.2$ | $20.8 \pm 0.4$ | $51.2 \pm 0.3$ | $42.2 \pm 0.4$ | $17.2 \pm 0.2$ |

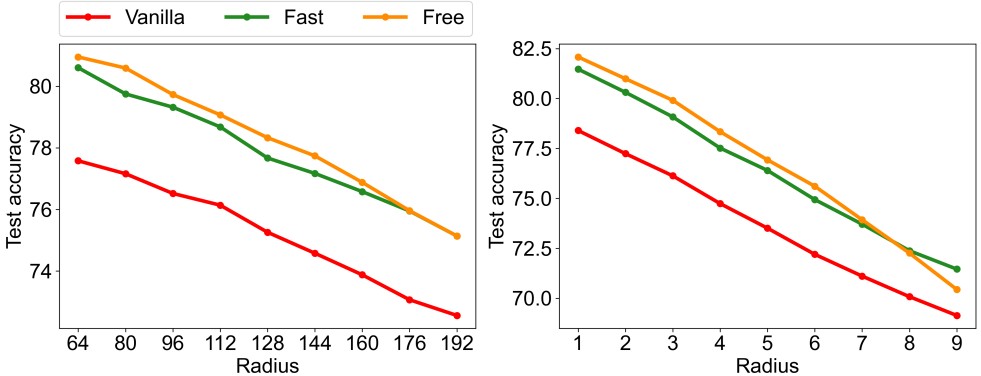

Figure 2: Robust accuracy of ResNet18 models adversarially trained by vanilla, fast, and free algorithms against square attack on CIFAR-10. The left figure applies $\mathcal{L}_2$ attacks of radius ranging from 64 to 192, and the right figure applies $\mathcal{L}_\infty$ attacks of radius ranging from 1 to 9.

**Robust Overfitting during Training Process:** We applied $\mathcal{L}_2$-norm attack of radius $\varepsilon = 128/255$ and $\mathcal{L}_\infty$-norm attack of radius $\varepsilon = 8/255$ to adversarially train ResNet18 models on CIFAR-10. For the vanilla algorithm, we used a PGD adversary with 10 iterations and step-size $\varepsilon/4$. For the fast algorithm, we used step-size $\tilde{\alpha}_\delta = \varepsilon/2$ for the $\mathcal{L}_2$-norm attack and $\tilde{\alpha}_\delta = \varepsilon$ for the $\mathcal{L}_\infty$-norm attack. For the free algorithm, we applied the learning rate of adversarial attack $\alpha_\delta = \varepsilon$ with free step $m = 4$. The other implementation details are deferred to Appendix B.1. We trained the models for 200 epochs and after every epoch, we tested the models' robust accuracy against a PGD adversary and evaluated the generalization gap. The numerical results are presented in Table 1. Also, the training curves are plotted in Figure 1.

Based on the empirical results, we observe the significant overfitting in the robust accuracy of the vanilla adversarial training: the generalization gap is above 30% against $\mathcal{L}_2$ attack and 50% against $\mathcal{L}_\infty$ attack. On the other hand, the free AT algorithm has less severe overfitting and reduced the generalization gap to 20%. Although the free AT algorithm applies a weaker adversary, it achieves comparable robustness on test samples to the vanilla AT algorithm against the PGD attacks by lowering the generalization gap. Additional numerical results for different numbers of free AT steps and on other datasets are provided in Appendix B.1.

**Robustness Evaluation Against Black-box Attacks:** To study the consequences of the generalization behavior of the free AT algorithm, we evaluated the robustness of the adversarially-trained networks against black-box attack schemes where the attacker does not have access to the parameters of the target models (Bhagoji et al., 2018). We applied the square attack (Andriushchenko et al., 2020), a score-based methodology via random search, to examine networks adversarially trained by the discussed algorithms as shown in Figure 2. We also used adversarial examples transferred from other independently trained robust models as shown in Figure 3. More experiments on different datasets are provided in Appendix B.2.

We extensively observe the improvements of the free algorithm compared to the vanilla algorithm against different black-box attacks, which suggests that its robustness is not gained from gradient-masking (Athalye et al., 2018) but rather attributed to the smaller generalization gap.

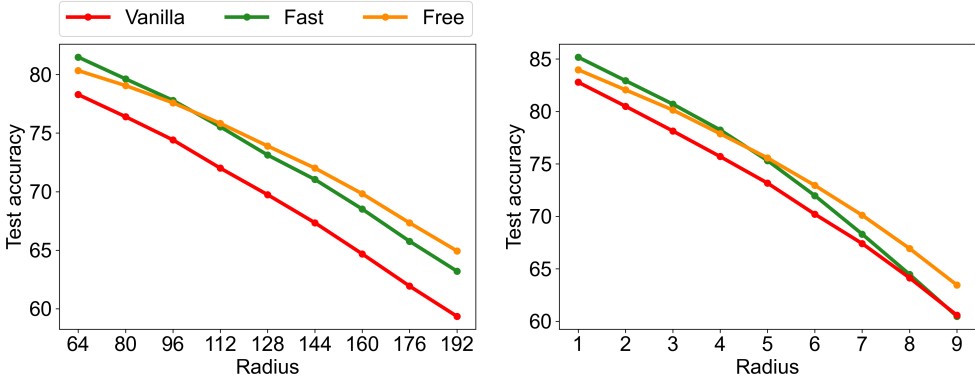

Figure 3: Robust accuracy against transferred attacks designed for another independently trained robust model. The left figure applies $\mathcal{L}_2$ attacks and the right figure applies $\mathcal{L}_\infty$ attacks.

**Generalization Gap for Different Numbers of Training Samples:** To examine our theoretical results in Theorems 2 and 4, we evaluated the robust generalization loss with respect to different numbers of training samples $n$. We randomly sampled a subset from the CIFAR-10 training dataset of size $n \in \{10000, 20000, 30000, 40000, 50000\}$, and adversarially trained ResNet18 models on the subset for a fixed number of iterations. As shown in Figure 4, the generalization gap of free AT is notably decreasing faster than vanilla AT with respect to $n$, which is consistent with our theoretical analysis. More experimental results are discussed in the Appendix B.3.

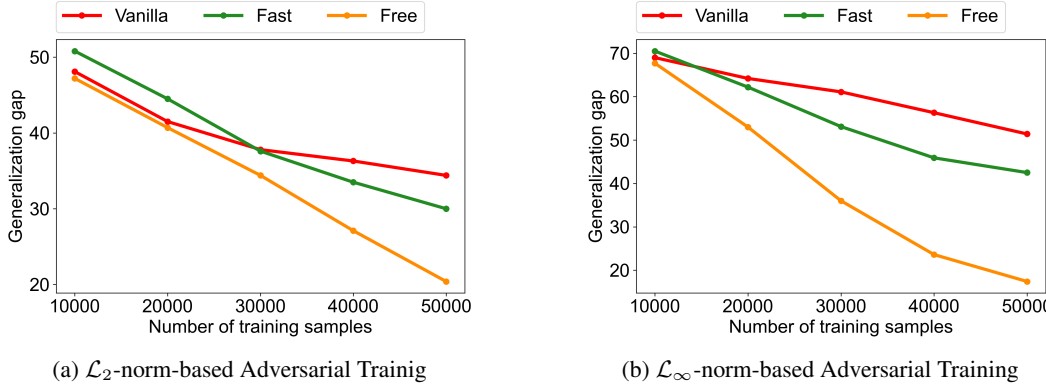

(a) $\mathcal{L}_2$-norm-based Adversarial Trainig

(b) $\mathcal{L}_\infty$-norm-based Adversarial Training

Figure 4: Adversarial generalization gap of ResNet18 models adversarially trained by vanilla, fast, and free algorithm for a fixed number of steps on a subset of CIFAR-10.

## 7 CONCLUSION

In this work, we studied the role of min-max optimization algorithms in the generalization performance of adversarial training methods. We focused on the widely-used free adversarial training method and, leveraging the algorithmic stability framework we compared its generalization behavior with that of vanilla adversarial training. Our generalization bounds suggest that not only can the free AT approach lead to a faster optimization compared to the vanilla AT, but also it can result in a lower generalization gap between the performance on training and test data. We note that our theoretical conclusions are based on the upper-bounds following from the algorithmic stability-based generalization analysis, and an interesting topic for future study is to prove a similar result for the actual generalization gap under simple linear or shallow neural net classifiers. Another future direction could be to extend our theoretical analysis of the simultaneous optimization updates to other adversarial training methods such as TRADES (Zhang et al., 2019) and ALP (Kannan et al., 2018).

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
