# OpenReview forum: "Stability and Generalization in Free Adversarial Training"
_ICLR.cc/2024/Conference — Submitted to ICLR 2024_

### Official Review · Reviewer_wev8 · 2023-10-19

**Soundness:** 3 good
**Presentation:** 3 good
**Contribution:** 3 good
**Rating:** 6
**Confidence:** 4

**Summary:**

This work utilizes the stability generalization framework to quantify the generalization bound of the Free Adversarial Training algorithm. Additionally, it shows that Free AT has a smaller generalization gap (but test error may not be smaller) and provides some theoretical intuitions.

**Strengths:**

This work offers important insights into the convergence properties of free AT. Additionally, the authors highlight intuitive relationships between "more simultaneous" gradient updates during training and the resulting generalization capability. These results can inspire further improvements in robust training algorithms to alleviate overfitting. The paper is generally well-presented and easy to follow.

**Weaknesses:**

- The theoretical support for FreeAT having a smaller generalization gap than VanillaAT could be more rigorous. Specifically, while Theorem 2 presents pessimistic results for the convergence of VanillaAT, it is unclear whether this bound is tight. It is unclear whether the convergence difference between vanilla and free AT is due to the algorithm itself or some artifacts of the proof technique. While experiment results support this intuition, the paper would benefit from some additional explanations. It would be even better if some lower bounds could be provided for $\mathcal{E}\_{\textrm{gen}} (A\_{\textrm{Vanilla}})$.
- The relationship between a smaller generalization gap and better transferability is unclear. The motivation for the experiment setting of transferring attacks from a robust model to a standard model is weak. I suggest moving the transferability analysis to the Appendix (it's still good to have them) and making space for Table 2, which supports your main claim.
- Figure 1 should use different line styles for train and test. In the current form, it's hard to distinguish them.
- Since the proposed convergence bounds depend on the dataset size $n$, this paper would benefit from some empirical comparisons between free and vanilla AT with different $n$ values.

I believe that the value of the theoretical bound on FreeAT's convergence outweighs the above weaknesses. Hence, a rating of 6 is given.

**Questions:**

- Do the theoretical results also apply to the $\ell_\infty$ case?
- Has there been any work that empirically estimates the Lipschitz and smoothness constants in Assumptions 1 and 2?
- Can smoothness and Lipschitzness assumptions (1 and 2) be relaxed? Specifically, instead of having this condition for all pairs of $\delta, \delta'$, is it possible to define Lipschitzness and smoothness over $\delta$ w.r.t. the nominal point ($\delta = 0$)? This relaxation will make the conditions more realistic.
- What is Free-4 in Table 2 and some of the figures (including Figure 1)?
- Theorem 3 lower-bounds $\mathbb{E}[ || w(S) - w(S') || ]$. However, does a large $\mathbb{E}[ || w(S) - w(S') || ]$ necessarily translate to large $\mathcal{E}\_{\textrm{gen}}$? Isn't it the case that neural networks with very different weights can have similar behavior?
- In practice, the attack loss function and the training loss function may not be the same, and using different losses has been empirically shown to decrease the generalization gap. Examples include TRADES [1] and ALP [2]. It's probably a stretch goal, but is it possible to extend the analysis to this scenario?

[1] Zhang, Hongyang, et al. "Theoretically principled trade-off between robustness and accuracy." International conference on machine learning. PMLR, 2019. \
[2] Harini Kannan, Alexey Kurakin, and Ian Goodfellow. "Adversarial logit pairing." arXiv preprint
arXiv:1803.06373, 2018.

---

> ### Author Response · Authors · 2023-11-18
> **Authors' Response to Reviewer wev8 (Part 1)**
>
> We thank Reviewer wev8 for his/her time and constructive feedback and suggestions. Below is our response to the questions and comments in the review.
>
> **1- Tightness of the generalization bound**
>
> **Re**: The reviewer is right that the generalization bound in Theorems 2 and 4 could be pessimistic in the sense that we only assume bounded Lipschitz and smoothness coefficients of the learning objective function. We would like to clarify that we did not make a definite theoretical statement on the generalization comparison of vanilla and free adversarial training, since our stability-based generalization bounds *only suggest* a lower expected generalization gap for free AT in comparison to vanilla AT. As pointed out by the reviewer, a generalization lower-bound would shed light on the tightness of the bounds in Theorems 2, 4. However, to the best of our knowledge, the question of a non-trivial lower-bound on the *expected* generalization gap (the quantity bounded in the stability-based approach) is still an open problem for both nonconvex-concave and nonconvex-nonconcave min-max learning problems.
>
> To investigate the discussed implications of our theoretical bounds, we performed numerical experiments on standard datasets and neural net architectures, which is similar to other theoretical works on the generalization of adversarial learning algorithms. The results of the numerical experiments are consistent with our hypothesis on the comparison of generalization error between vanilla and free AT algorithms.
>
> **2- Postponing the numerical results on transferability to the Appendix**
>
> **Re**: As recommended by the reviewer, we deferred the numerical results on the attack transferability to the Appendix, and used the space to present the numerical results on the role of the size of training set $n$, which was suggested by the reviewer.
>
> **3- Line styles in Figure 1 for train and test qunatities**
>
> **Re**: We thank the reviewer for pointing out the identical line styles in Figure 1. We have changed the line style for test accuracy in the revision.
>
> **4- Empirical comparison between free and vanilla AT with different dataset size $n$ values**
>
> **Re**: Following the reviewer’s suggestion, we performed numerical experiments by randomly sampling a subset of size $n \in \lbrace 10000, 20000, 30000, 40000, 50000\rbrace$ from CIFAR-10 and CIFAR-100 training sets, and trained ResNet18 neural nets using those training sets. The numerical results of our comparison of the generalization gaps between free and vanilla AT are presented in Figure 4 and Appendix B.3 of the revised submission. The results indicate that the generalization gap of free AT is decreasing considerably faster than vanilla AT with dataset size $n$. We thank the reviewer for the great suggestion.
>
> **5- “Do the theoretical results also apply to the $L_{\infty}$ case?”**
>
> **Re**: Our theoretical analysis concerns only the $L_2$-norm bounded perturbations. As a Hilbert-space norm induced by an inner product, the $L_2$-based projection reduces to a standard normalization of the input vector which preserves the vector’s direction. This property allows us to leverage the theory of linear dynamical systems to track the changes following the two different datasets $S,S’$ and bound the stability degree of AT algorithms. On the other hand, the $L_\infty$-norm would lead to a sign-based projection that can completely alter the direction of an input vector. Therefore, a similar stability analysis of the $L_\infty$-based AT would lead to a highly non-linear dynamical system which would be difficult to analyze using existing control and optimization theory frameworks.
>
> **6- "Has there been any work that empirically estimates the Lipschitz and smoothness constants in Assumptions 1 and 2?"**
>
> **Re**: To the best of our knowledge, computing the exact Lipschitz and smoothness coefficients of overparameterized neural networks is a computationally challenging problem. The related works [3,4] used SDP relaxation to compute the Lipschitz constant of neural networks, but the computational cost of a SDP-based method would be unaffordable for overparameterized neural networks such as standard ResNets. Another line of work [5] attempted to bound the Lipschitz constants and smoothness by using standard matrix-based norm for the neural net’s weight matrices; however, these upper-bounds could be significantly loose in application to standard neural net architectures and image datasets.
>
> **7- Relaxation of smoothness and Lipschitzness assumptions**
>
> **Re**: We note that our stability-based generalization results will hold if the bounded gradient and Hessian norm assumptions hold only for every input vector within an $\epsilon$-distance from the support set of the underlying distribution of random vector $X$. We have clarified this point in the revised text.

---

> ### Author Response · Authors · 2023-11-18
> **Authors' Response to Reviewer wev8 (Part 2)**
>
> **8- "What is Free-4 in Table 2 and some of the figures?"**
>
> **Re**: We added footnote 1 for clarity. Throughout our work, ‘‘Free-$m$’’ means the free AT algorithm with $m$ free steps, and following from [8] and [9], we use ‘‘Free’’ without specification to denote Free-4 by default.
>
> **9- Relationship between $\mathbb{E}[|| w(S) - w(S’) ||]$ and $\mathcal{E}_{gen}$**
>
> **Re**: We agree with the reviewer that a higher stability degree does not necessarily imply a lower generalization gap. Our generalization analysis follows the standard analysis for stochastic gradient-based optimization algorithms (please see the discussion in [7] for more details), and, as the reviewer pointed out, can be pessimistic for a general nonconvex-nonconcave minimax problem. Similar to other theoretical works on generalization of deep learning models, our bounds only suggest a hypothesis for the generalization comparison, which we attempt to validate using numerical experiments.
>
> **10- Extension of the analysis to other AT algorithms**
>
> **Re**: We thank the reviewer for the constructive feedback. In Appendix B.6 of our revision, we have performed numerical experiments on a similar ”free” variant of the TRADES algorithm [6] (which we call Free-TRADES). Following our theoretical discussion, the simultaneous optimization updates in Free-TRADES could help lower the generalization gap. Our numerical results indicate that Free-TRADES can also reduce the generalization error. The theoretical analysis of the generalization behavior of Free-TRADES is an interesting direction for future exploration.
>
>
> [1] Yue Xing, Qifan Song, and Guang Cheng. On the algorithmic stability of adversarial training. Advances in neural information processing systems, 2021.
>
> [2] Jiancong Xiao, Yanbo Fan, Ruoyu Sun, Jue Wang, and Zhi-Quan Luo. Stability analysis and generalization bounds of adversarial training. Advances in Neural Information Processing Systems, 2022b.
>
> [3] Mahyar Fazlyab, Alexander Robey, Hamed Hassani, Manfred Morari, and George Pappas. Efficient and accurate estimation of lipschitz constants for deep neural networks. Advances in Neural Information Processing Systems, 2019.
>
> [4] Zhouxing Shi, Yihan Wang, Huan Zhang, J Zico Kolter, and Cho-Jui Hsieh. Efficiently computing local lipschitz constants of neural networks via bound propagation. Advances in Neural Information Processing Systems, 2022.
>
> [5] Bohang Zhang, Du Jiang, Di He, and Liwei Wang. Rethinking lipschitz neural networks and certified robustness: A boolean function perspective. Advances in Neural Information Processing Systems, 2022.
>
> [6] Hongyang Zhang, Yaodong Yu, Jiantao Jiao, Eric Xing, Laurent El Ghaoui, and Michael Jordan. Theoretically principled trade-off between robustness and accuracy. In International Conference on Machine Learning, 2015.
>
> [7] Moritz Hardt, Benjamin Recht, and Yoram Singer. Train faster, generalize better: Stability of stochastic gradient descent. arXiv preprint arXiv:1509.01240, 2015.
>
> [8] Ali Shafahi, Mahyar Najibi, Mohammad Amin Ghiasi, Zheng Xu, John Dickerson, Christoph Studer, Larry S Davis, Gavin Taylor, and Tom Goldstein. Adversarial training for free, Advances in Neural Information Processing Systems, 2019.
>
> [9] Eric Wong, Leslie Rice, and J Zico Kolter. Fast is better than free: Revisiting adversarial training. International Conference on Machine Learning, 2020.

---

> > ### Comment · Reviewer_wev8 · 2023-11-19
> > **Thank you for the response.**
> >
> > Thank you for the response, which clarified most of my questions.
> >
> > This paper presents an interesting and sound analysis, with plenty of empirical evidence after the revision. Therefore, I believe that it is clearly *above the acceptance threshold*.
> >
> > That being said, some of the theoretical results are less practical and meaningful (such as the relationship between $\mathbb{E} [|| w(S) - w(S') ||]$ and $\mathcal{E}_{gen}$) due to the technical difficulties. As a result, I am keeping my rating of 6.

---

> > > ### Author Response · Authors · 2023-11-23
> > > **Thank you for your feedback**
> > >
> > > We thank Reviewer wev8 for his/her time and feedback on our response.

---

### Official Review · Reviewer_T7Bj · 2023-10-20

**Soundness:** 2 fair
**Presentation:** 3 good
**Contribution:** 2 fair
**Rating:** 3
**Confidence:** 4

**Summary:**

The paper studies the generalization of free adversarial training (AT) which was proposed in Shafahi et al. (2019).
 The authors use the algorithmic stability approach to analyze its generalization behavior and it provides its comparison of the generalization bounds against the vanilla, fast AT methods.  It claims that the free AT algorithm could have a lower generalization bound than the vanilla AT one.

**Strengths:**

This seems to be the first-ever-known result that addressed the generalization of the free AT method using the algorithmic stability approach in the setting of mini-max formulation.

**Weaknesses:**

While the stability results for the free AT method are first-ever-known, the proof techniques seem to be incremental and the paper did not illustrate clearly what the main technical contribution is, particularly considering there is a considerable amount of work on stability analysis.

The generalization bound in Theorem 4 relies on the restrictive assumption that the gradient $\nabla_\delta h(w,\delta; x,y)$ is lower-bounded by $1 / \psi$ during the training process.  There is no discussion about when this critical condition holds true.

**Questions:**

It is not clear to me what the free AT method aims to minimize or optimize.  The objective function of the vanilla AT method is given on page 3, i.e. $R_S(w)$ or $R(w)$.  From the pseudo-code of Algorithm 3,  there are two random samplings--one for mini-batch and one for $\{\delta_j\}$ and then the $w$ and $\delta$ are updated by gradient descent and ascent, respectively.  In this sense, does the free AT methods aim to minimize the following objective
$$ \min_w \max_\delta {1\over n } \sum_{j=1}^n \int_{\delta_j\in \Delta} h(w,\delta_j; x_j,y_j)$$
The objective functions seem to be very different from each other for the free AT method and the vanilla AT one.   Indeed, the objective function of the free AT method is a low-bound relaxation of the vanilla one.   Could you explain more about this point?

---

> ### Author Response · Authors · 2023-11-18
> **Authors' Response to Reviewer T7Bj**
>
> We thank Reviewer T7Bj for his/her time and feedback. Below is our response to the questions and comments in the review.
>
> **1- Technical contributions of the work**
>
> **Re**: Free AT is a widely used variant of adversarial training, mainly due to its higher speed in application to large-scale learning problems. While the free AT method has been extensively used in many applications, the existing theory literature on adversarial training methods has not focused on the specific properties of free AT algorithm. In our work, we focus on the generalization analysis of models learned by free AT, which, to the best of our knowledge, has not been studied in previous works.
>
> Regarding the reviewer’s comment on our work’s technical contribution, we note that our generalization guarantee for free AT (Theorem 4) does not follow from any existing stability-based generalization bound for min-max learning frameworks. Specifically, our generalization analysis accounts for the following properties of the free AT method, which have not been considered in the related works:
>
> 1- Re-initialization of the max variable (adversarial perturbations) after the completion of optimization steps for every batch of training data\
> 2- Utilizing the normalized gradient (instead of vanilla gradient) for the gradient ascent step of solving the maximization sub-problem\
> 3- Using mini-batch stochastic optimization for updating min and max variables at every iteration
>
> The generalization analysis under the above conditions and the comparison of generalization performance between vanilla vs. free adversarial training are our work’s main technical contributions, which we believe are novel results that do not follow from any existing bound in the literature.
>
> **2- The assumption on bounded gradient norm $\Vert\nabla_\delta h(w,\delta;x,y)\Vert$ in Theorem 4**
>
> **Re**: We note that the bounded-gradient-norm assumption is only needed for the points within an $\varepsilon$-distance from the training samples. Based on this comment, we numerically evaluated the gradient norm in the application of free-AT to CIFAR-10 and CIFAR-100 datasets, indicating that the minimum gradient norm on training data is constantly lower-bounded by $\mathcal{O}(10^{-3})$ in those experiments. We refer the reviewer to Figure 11 in the revised Appendix B.5 for the statistics of the gradient norm evaluation in the experiments.
>
> **3- The objective that the free AT method aims to optimize**
>
> **Re**: The free AT algorithm has been originally proposed in [3] to solve the standard min-max formulation of adversarial training problems. The reviewer’s question points to an interesting research direction on the effects of the optimization steps in free AT on the target min-max optimization problem. While this problem sounds an interesting direction to explore on free AT, we think our stability-based generalization analysis is orthogonal to this research direction.
>
> Please note that our main objective is to compare the generalization error of free vs. vanilla AT methods, and to perform a fair comparison we should use the same generalization error metric for both these AT algorithms. According to the standards in the adversarial learning literature, we chose this generalization metric to be the difference between the worst-case training and test error under norm-bounded perturbations. Therefore, to ensure a fair comparison between free and vanilla adversarial training, we have to use the same generalization metric for the analysis of free AT. Any different selection of the generalization metric for free AT would have led to an unfair comparison of the generalization error between the two methods.
>
>
> [1] Yue Xing, Qifan Song, and Guang Cheng. On the algorithmic stability of adversarial training. Advances in neural information processing systems, 2021.
>
> [2] Jiancong Xiao, Yanbo Fan, Ruoyu Sun, Jue Wang, and Zhi-Quan Luo. Stability analysis and generalization bounds of adversarial training. Advances in Neural Information Processing Systems, 2022.
>
> [3] Ali Shafahi, Mahyar Najibi, Mohammad Amin Ghiasi, Zheng Xu, John Dickerson, Christoph Studer, Larry S Davis, Gavin Taylor, and Tom Goldstein. Adversarial training for free, Advances in Neural Information Processing Systems, 2019.

---

### Official Review · Reviewer_ge9g · 2023-10-29

**Soundness:** 2 fair
**Presentation:** 2 fair
**Contribution:** 2 fair
**Rating:** 6
**Confidence:** 3

**Summary:**

This work studies the role of min-max optimization algorithms in the generalization performance of adversarial training methods. It leverages the algorithmic stability framework to compare the generalization behavior of adversarial training methods. The developed generalization bounds suggest that not only can the free AT approach lead to a faster optimization compared to the vanilla AT, but also it can result in a lower generalization gap between the performance on training and test data.

**Strengths:**

- This work provides some theoretical results.
- The theoretical conclusions are easy to follow.

**Weaknesses:**

- What is the definition of $\Delta$?
- What is the definition of randomized algorithm $A(\cdot)$? A mapping? If yes, then what is the definition of $\mathbb{E}_A$?
- Given $S$, $w=A(S)$ is a random variable or constant？
- What's the definition of $S'$ here?
- What is the definition of "$A$ is $\epsilon$-uniformly stable"?
- Unclear definition in Theorem 1?
- I **guess** the theory is developed over "randomized algorithm $A$" (and Gibbs loss?), i.e., the output of $A(S)$ is random weights, which is distributed by a posterior. However, this paper only presents empirical results based on deterministic weights. How can these empirical findings provide support for the theoretical results?
- If $A(S)$ is a random variable, given $S$, what are the specific posterior distributions of $A_{AVanilla}(S)$ and $A_{Free}(S)$? What is the difference between these posterior distributions? Where does the randomness of $A_{AVanilla}(S)$ come from?
- It seems there are not some interesting insights from the theoretical and empirical results in this work.

(Please correct me if I have some mistakes.)

**Questions:**

See weakness.

---

> ### Author Response · Authors · 2023-11-18
> **Author Response to Reviewer ge9g**
>
> We thank Reviewer ge9g for his/her time and feedback. Below is our response to the questions and comments in the review.
>
> **1- “What is the definition of $\Delta$?”**
>
> **Re**: As we stated at the beginning of Section 3.1, $\Delta$ is the set of all possible perturbations, which is usually an $L_2$-norm or $L_\infty$-norm bounded ball of some radius $\varepsilon$.
>
> **2- “What is the definition of randomized algorithm $A(\cdot)$? What is the definition of $\mathbb{E}_A$? Given $S$, $w=A(S)$ is a random variable or constant?”**
>
> **Re**: As we stated in Section 3, $A$ is a potentially randomized algorithm that takes a dataset $S$ as input and outputs model parameter $w=A(S)$. Since the randomness of $A$ may come from random initialization or random batch selection, $\mathbb{E}_A$ is taking the expectation over such randomness, and $w=A(S)$ will therefore be a random vector.
>
> **3- “What's the definition of $S’$ here?”**
>
> **Re**: Following our statement in Definition 1, $S’$ is any dataset that has the same size as $S$ and differs from $S$ in at most one example.
>
> **4- “What is the definition of "$A$ is $\epsilon$-uniformly stable"?”**
>
> **Re**: Please refer to Definition 1 for the formal definition of “$\epsilon$-uniformly stable”, which follows from references [1,2].
>
> **5- “Unclear definition in Theorem 1?”**
>
> **Re**: We wish the reviewer could be more specific about his/her observed unclarities in Theorem 1. We will be happy to answer any follow-up question on this point.
>
> **6- “Where does the randomness of $A_{Vanilla}(S)$ come from?”**
>
> **Re**: In our analysis, both $A_{Vanilla}$ and $A_{Free}$ are stochastic optimization algorithms and the training dataset is also random. So the randomness of $A$ could come from both random initialization and random batch selection.
>
>
> **7- Insights from the theoretical and empirical results.**
>
> **Re**: Free AT is a widely used variant of adversarial training methods due to its higher speed in training the neural net classifier. While the free AT method has been extensively used in several applications, the existing theoretical analysis of adversarial training have not focused on the properties of free AT algorithm. In our work, we focus on the generalization analysis of models learned by the free AT approach, which, to the best of our knowledge, has not been studied in the literature. We show that due to the simultaneous updating nature of free AT, it has significantly better generalization performance compared to vanilla AT in theoretical analysis, and the numerical results are consistent with the theory. Please see our response to the first comment of Reviewer T7Bj for more information about our technical contributions.
>
> [1] Yue Xing, Qifan Song, and Guang Cheng. On the algorithmic stability of adversarial training. Advances in neural information processing systems, 2021.
>
> [2] Jiancong Xiao, Yanbo Fan, Ruoyu Sun, Jue Wang, and Zhi-Quan Luo. Stability analysis and generalization bounds of adversarial training. Advances in Neural Information Processing Systems.

---

> > ### Comment · Reviewer_ge9g · 2023-11-18
> > **response**
> >
> > Dear Authors,
> >
> > Thank you for your response.
> >
> > After reviewing your clarifications, I have a better understanding of the manuscript. However, I still believe that clear definitions could benefit from further elucidation. For example, $\Delta$ is introduced early in Section 3, but its definition appears later. It would be better to explicitly define what is a randomized algorithm. Does this include all forms of randomness during training, such as dropout, all initial weight distributions, and so on? Does the bound hold under any randomness of training?
> >
> > According to the rebuttal, I have adjusted my score to 6 and decreased my confidence to 3. All in all, **I stand on the fence for this manuscript**.
> >
> > Reviewer ge9g

---

> > > ### Author Response · Authors · 2023-11-23
> > > **Thank you for your feedback**
> > >
> > > We thank Reviewer ge9g for his/her feedback on our response. We have further clarified the definition of perturbation set $\Delta$ and randomized algorithm $A$ in the revision. We note that our definition of a randomized learning algorithm is consistent with the literature on algorithmic stability-based generalization analysis (e.g. see the references [1,2,3]) and captures any source of randomness in the algorithm, e.g. stochastic batch selection and random initialization of weights in stochastic gradient methods. We will be happy to respond to any remaining questions of the reviewer.
> > >
> > >
> > >
> > > [1] Moritz Hardt, Benjamin Recht, and Yoram Singer. Train faster, generalize better: Stability of stochastic gradient descent. International Conference on Machine Learning, 2015.\
> > > [2] Yue Xing, Qifan Song, and Guang Cheng. On the algorithmic stability of adversarial training. Advances in neural information processing systems, 2021.\
> > > [3] Jiancong Xiao, Yanbo Fan, Ruoyu Sun, Jue Wang, and Zhi-Quan Luo. Stability analysis and generalization bounds of adversarial training. Advances in Neural Information Processing Systems, 2022.

---

### Official Review · Reviewer_iGFe · 2023-10-30

**Soundness:** 3 good
**Presentation:** 3 good
**Contribution:** 2 fair
**Rating:** 6
**Confidence:** 4

**Summary:**

This paper studies stability and generalization of vanilla, free, and fast adversarial training from an algorithmic stability perspective. The generalization error gap bounds are derived for those adversarial training methods, and numerical results are also provided to show the generalization performance and robustness against black-box attacks.

**Strengths:**

This paper is well-motivated and well-written. The novelty and contributions are clearly stated and organized. The theoretical findings are provided in a rigorous manner, together with some validation numerical results. In general, the theoretical findings are interesting to the community.

**Weaknesses:**

1. This paper is dedicated to generalization performance analysis of existing adversarial training methods and reveals some interesting points. Nevertheless, there is a lack of deep insights on the new advanced designs of adversarial training from the generalization bounds. The authors should have discussed the insights/guidance from the theoretical findings, or discussed certain limitations of the algorithmic stability approach itself.
2. From the experimental results, e.g., Figure 1, it appears that the reduced generalization error gap of free adversarial training is mainly due to the higher training error. Assuming the generalization error gap maintains, it is unclear if the test error can be further reduced when the training error is reduced. The authors should add some comments on this.
3. It would expect that new training/regularization methods could be proposed given the obtained generalization error bounds. Otherwise, the impact of the theoretical findings of this work is quite limited. A thorough discussion would be helpful and beneficial. It would be also interesting to know the potential connection between generalization gap and the robustness against adversarial attacks.

**Questions:**

See the Weaknesses above.

Add some comments on the practical usefulness of the theoretical findings with respect to the design of adversarial training methods. The limitations of the algorithmic stability approach for studying generalization performance could be also discussed.

---

> ### Author Response · Authors · 2023-11-18
> **Authors' Response to Reviewer iGFe**
>
> We thank Reviewer iGFe for his/her time and constructive feedback. Below is our response to the questions and comments in the review.
>
> **1- Practical implications of the theoretical generalization bounds**
>
> **Re**: In the revised version, we have added a remark after Theorem 4 to discuss the implications of the theoretical bound on Free AT. We note that Theorem 4 suggests how the stepsize parameters for the max and min variables affect the generalization gap of free AT, where by reducing the number of maximization steps $m$ and step size $\alpha_\delta$ one can achieve a lower generalization error.
>
>
> **2- The relationship between generalization gap and training error**
>
> **Re**: We agree with the reviewer that if we continue training the neural networks by applying free AT for more epochs, the training error might further decrease and the generalization gap might be larger. This phenomenon is essentially consistent with the stability analysis – our stability-based generalization bounds consider the number of iterations $T$ as a parameter and grow sublinearly with $T$. To make a fair comparison between free and vanilla AT from the stability perspective, we use the same number of training iterations in our experiments, and we observe a lower generalization gap and sometimes 2%-5% improvement on the robust test accuracy by free AT. Please note that the dependence of the generalization gap on the iteration number $T$ is consistent with the numerical findings of the references (e.g. the discussion in [1]).
>
>
> **3- Improved adversarial training/regularization methods based on the generalization analysis**
>
> **Re**: The simultaneous nature of min-max updates of free AT can be extended to other established robust training methods such as TRADES [2]. Based on this comment, we have performed numerical experiments on the application of a similarly defined ‘‘free’’ version of TRADES (which we call Free-TRADES). The results of our numerical experiments are presented in the revised Appendix B.6. The numerical results indicate that Free-TRADES can similarly gain a better generalization performance using the simultaneous min and max optimization steps. The theoretical analysis of the proposed Free–TRADES will be an interesting future direction to our work, which we have discussed in the revised conclusion section.
>
> [1] Leslie Rice, Eric Wong, and Zico Kolter. Overfitting in adversarially robust deep learning. In International Conference on Machine Learning, 2020.
>
> [2] Hongyang Zhang, Yaodong Yu, Jiantao Jiao, Eric Xing, Laurent El Ghaoui, and Michael Jordan. Theoretically principled trade-off between robustness and accuracy. In International conference on machine learning, 2019.

---

### Meta-Review · Area_Chair_H6sr · 2023-12-15

**Metareview:**

This paper analyzes and compares the generalization performance of some adversarial training methods using an algorithmic stability framework. Three reviewers have found the work valuable while another reviewer has been highly critical of its contribution. There are issues with the presentation of the paper, which have been adequately responded by the authors. The major concern is about whether the results of this paper offer any new insight or have any practical implications. The paper relies on technical assumptions that could be potentially very restrictive.

Based on the recommendation by the SAC, we have decided to reject the paper.

**Justification For Why Not Higher Score:**

N/A

**Justification For Why Not Lower Score:**

N/A

---

### Decision · Program_Chairs · 2024-01-16

Reject